# EQUILIBRIUM MATCHING: GENERATIVE MODELING WITH IMPLICIT ENERGY-BASED MODELS

## ABSTRACT

We introduce *Equilibrium Matching* (EqM), a generative modeling framework built from an equilibrium dynamics perspective. EqM discards the non-equilibrium, time-conditional dynamics in traditional diffusion and flow-based generative models and instead learns the equilibrium gradient of an implicit energy landscape. Through this approach, we can adopt an optimization-based sampling process at inference time, where samples are obtained by gradient descent on the learned landscape with adjustable step sizes, adaptive optimizers, and adaptive compute. EqM surpasses the generation performance of diffusion/flow models empirically, achieving an FID of 1.90 on ImageNet 256×256. EqM is also theoretically justified to learn and sample from the data manifold. Beyond generation, EqM is a flexible framework that naturally handles tasks including partially noised image denoising, OOD detection, and image composition. By replacing time-conditional velocities with a unified equilibrium landscape, EqM offers a tighter bridge between flow and energy-based models and a simple route to optimization-driven inference.

## 1 INTRODUCTION

Generative modeling has advanced rapidly with diffusion and flow-based methods (Sohl-Dickstein et al., 2015; Ho et al., 2020; Song et al.; Lipman et al.; Liu et al.), which map simple noise distributions to complex data by defining a forward noising process and learning its reverse. While they achieve state-of-the-art sample quality (Nichol and Dhariwal, 2021; Dhariwal and Nichol, 2021; Karras et al., 2022), these models employ non-equilibrium dynamics at both training and inference. Diffusion/flow models are conditioned on input timestep and learn distinct dynamics for inputs at different noise levels. This non-equilibrium design imposes practical constraints such as noise level schedule and fixed integration horizon during sampling.

Existing approaches that attempt to learn equilibrium dynamics suffer from different problems. Sun et al. (2025) have shown that forcing diffusion models to learn equilibrium dynamics by simply removing time (noise[1]) conditioning leads to worse generation quality. Energy-based models (EBMs) (LeCun et al., 2006; Du and Mordatch, 2019; Carreira-Perpinan and Hinton, 2005) directly learn equilibrium energy landscapes, but they often suffer from training instabilities (Du et al., 2020b) and poor sample quality (Du and Mordatch, 2019; Nijkamp et al., 2020). More recent approaches such as Energy Matching (Balcerak et al., 2025) involve separate training stages and fail to surpass the generation quality of flow-based method on large-scale datasets.

In this work, we introduce *Equilibrium Matching* (EqM), a generative modeling framework from an equilibrium perspective. Equilibrium Matching replaces the time-conditional non-equilibrium dynamics of diffusion/flow models with a single time-invariant equilibrium gradient over an implicit energy landscape. We hypothesize that the quality degradation in noise-unconditional diffusion models originates from the incompatibility between the target gradient and equilibrium dynamics. To this end, we introduce a new family of target gradients that align with an implicit energy function. We also provide model variants that explicitly learn this energy function.

By learning a single equilibrium dynamics, Equilibrium Matching supports *optimization-based sampling*, where samples are obtained by gradient descent on the learned landscape. Unlike existing diffusion samplers that integrate along a prescribed trajectory, optimization-based sampling supports

---

[1]In the context of this work, time conditioning and noise conditioning are equivalent.

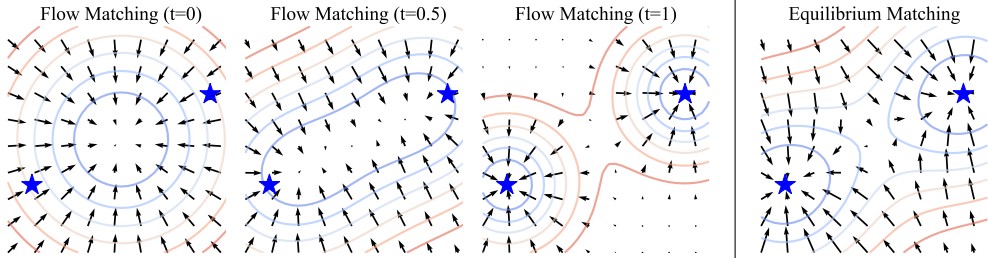

Figure 1: **Conceptual 2D Visualization.** We compare the conceptual 2D dynamics of Equilibrium Matching and Flow Matching under 2 ground truths (marked by stars). **Left**: Flow Matching learns *non-equilibrium* velocity that *varies* over time. **Right**: Equilibrium Matching learns an *equilibrium* gradient that is time-invariant.

different step sizes and adaptive optimizers. Existing gradient optimization techniques such as Nesterov Accelerated Gradient can be naturally adopted to achieve better generation quality. Equilibrium Matching can also allocate inference-time compute adaptively. It adjusts the sampling steps for each sample independently based on gradient norm and can save up to 60% of function evaluations.

We validate Equilibrium Matching through both theoretical analysis and empirical evidence. Theoretically, we show that Equilibrium Matching is guaranteed to learn the data manifold and produce samples from this manifold using gradient descent. Empirically, Equilibrium Matching achieves 1.90 FID on ImageNet $256\times256$ generation, outperforming existing diffusion and flow-based counterparts in generation quality. Equilibrium Matching also exhibits strong scaling behavior, exceeding the flow-based counterpart at all tested scales. These results suggest that Equilibrium Matching is a promising alternative for generative modeling.

Beyond generation, Equilibrium Matching demonstrates unique properties that traditional diffusion/flow-based models lack. Equilibrium Matching can generate high-quality samples directly from partially noised inputs, whereas flow-based models only perform well when starting with pure noise. Moreover, Equilibrium Matching can perform out-of-distribution (OOD) detection without relying on any external module. We also show that different Equilibrium Matching models can be added together to generate compositional images in a similar way as EBMs. Our results show that Equilibrium Matching offers capabilities unseen in traditional diffusion/flow models. We hope that Equilibrium Matching provides a principled way to unify flow-based and energy-based perspectives and can enable new inference-time strategies in the future.

## 2 PRELIMINARIES: FLOW MATCHING

Diffusion and Flow Matching models learn non-equilibrium dynamics. Flow Matching (FM), for example, learns to match the conditional velocity along a linear path connecting noise and image samples. During sampling, Flow Matching starts from pure Gaussian noise and iteratively denoises the current sample using the velocity predicted by $f$. This process is governed by a differential equation framework, in which the predicted velocity is treated as the time derivative of the desired sampling path and integrated over a total length of $1$. Formally, let $f$ denote the generative model, let $\epsilon$ be Gaussian noise, let $x$ be a real image sample from the training dataset, and let $t$ be a timestep sampled uniformly between $0$ and $1$. The training objective of FM can be written as:

$$L_{\text{FM}} = \big(f(x_t, t) - (x - \epsilon)\big)^2. \tag{1}$$

Here $x - \epsilon$ is the target velocity. The model $f$ takes both $x_t$ and $t$ as inputs, where $t$ is the corresponding noise level for $x_t$.

**Noise-Unconditional Model.** One approach to make Flow Matching learn equilibrium dynamics is to directly remove $t$ as a conditioning input for $f$ (Sun et al., 2025). In this noise-unconditional version, the model is no longer conditioned on the timestep or noise level, giving a new objective:

$$L_{\text{uncond-FM}} = \big(f(x_t) - (x - \epsilon)\big)^2. \tag{2}$$

The only difference is the absence of conditioning on $t$ in $f$. The sampling and training procedures are otherwise identical to those of the original Flow Matching. However, removing noise conditioning like the above degrades generation quality, and this unconditional variant does not exhibit unique properties that differ from those of the original model.

## 3 EQUILIBRIUM MATCHING

Equilibrium Matching (EqM) learns a time-invariant gradient field that is compatible with an underlying energy function, eliminating time/noise conditioning and fixed-horizon integrators. Conceptually (Fig. 1), EqM's gradient vanishes on the data manifold and increases toward noise, yielding an equilibrium landscape in which ground-truth samples are stationary points. We illustrate the difference between Equilibrium Matching and Flow Matching in Fig. 1. Flow Matching learns a varying velocity that only converges to ground truths at the final timestep, whereas EqM learns a time-invariant gradient landscape that always converges to ground-truth data points.

### 3.1 TRAINING

To train an Equilibrium Matching model, we aim to construct an energy landscape in which the target gradient at ground-truth samples is zero. To do so, we first define a corruption scheme that provides a transition between data and noise. Let $\gamma$ be an interpolation factor sampled uniformly between $0$ and $1$, let $\epsilon$ be Gaussian noise, and let $x$ be a sample from the training set. Denote $x_\gamma = \gamma x + (1 - \gamma)\epsilon$ as an intermediate corrupted sample. Unlike $t$ in FM, our $\gamma$ is implicit and not seen by the model. Our goal is to define a target gradient at these intermediate samples $x_\gamma$ that matches an implicit energy landscape. Using a gradient direction pointing from noise to data, we write the Equilibrium Matching training objective as:

$$L_{\text{EqM}} = \big(f(x_\gamma) - (\epsilon - x)c(\gamma)\big)^2, \tag{3}$$

where $c(\gamma)$ controls the gradient magnitude. The target gradient $(\epsilon - x)c(\gamma)$ has direction $(\epsilon - x)$ that points from noise to data and magnitude $c(\gamma)$ that vanishes as we get closer to the data manifold. We explicitly enforce $c(1) = 0$, ensuring that the energy landscape has vanishing gradients at real samples. Together, the direction and magnitude result in a target gradient that supports an implicit energy landscape. When $c(\gamma) = 1$, the EqM objective is exactly the negation of FM's objective.

Compared with Flow Matching, EqM's objective is derived from an EBM perspective rather than a normalizing flow's perspective. This results in a different direction of target, where EqM learns the gradient $\epsilon - x$ and FM learns the velocity $x - \epsilon$. The difference in perspectives also results in different fundamental constraints. EqM requires $c(1) = 0$ to construct an energy landscape with local minima at the data manifold, whereas FM requires $\int_{\gamma=0}^{1} c(\gamma) = 1$ to construct a valid integration path. Next, we investigate several simple choices for $c$ in EqM.

**Linear Decay.** A natural choice for $c$ is a linear function that decays from $1$ to $0$. In the energy landscape, this is equivalent to assigning noise a high gradient and making the gradient decay linearly to $0$ toward the ground-truth image:

$$c_{\text{linear}}(\gamma) = 1 - \gamma. \tag{4}$$

**Truncated Decay.** Beyond linear decay, we may want the gradient to remain constant when far away from data. This leads to a truncated decay, where the target gradient stays at $1$ when $\gamma \leq a$ ($a \in [0, 1)$) and then decays linearly to $0$ when approaching the data manifold:

$$c_{\text{trunc}}(\gamma) = \begin{cases} 1, & \gamma \leq a \\ \frac{1-\gamma}{1-a}, & \gamma > a \end{cases}. \tag{5}$$

**Piecewise.** We can also vary the constant segment of the truncated decay function and set its starting value to $b$, with $b \in [0, \infty)$. This gives a piecewise function that starts at $b$, decays linearly down to $1$ at $\gamma = a$, and then decays linearly down to $0$ at $\gamma = 1$:

$$c_{\text{piece}}(\gamma) = \begin{cases} b - \frac{b-1}{a}\gamma, & \gamma \leq a \\ \frac{1-\gamma}{1-a}, & \gamma > a \end{cases}. \tag{6}$$

**Gradient Multiplier.** The above choices for $c$ have varying magnitudes. Thus, we introduce an additional gradient multiplier $\lambda$ on top of these gradient fields to control the overall scale. Using linear decay as an example, the final function becomes $c(\gamma) = \lambda c_{\text{linear}}(\gamma) = \lambda(1 - \gamma)$.

## 3.2 LEARNING EXPLICIT ENERGY

Previously, we treat the energy landscape as an implicit underlying structure and learned the gradient of this implicit energy function. We can also modify the Equilibrium Matching model to learn explicit energy values. For an explicit energy model $g$, we match the gradient $\nabla g(x_\gamma)$ at corrupted samples and take $E = g(x_\gamma)$ as the energy at $x_\gamma$. This naturally constructs an energy landscape in which real samples are assigned low energy while noises are assigned high energy. The Equilibrium Matching with Explicit Energy (EqM-E) objective can be written as:

$$L_{\text{EqM-E}} = (\nabla g(x_\gamma) - (\epsilon - x)c(\gamma))^2. \tag{7}$$

$\nabla g(x_\gamma)$ is the derivative of $g$ with respect to input $x_\gamma$. To obtain a scalar function $g$, we follow Du et al. (2023) and discuss two different ways to construct $g$ from an existing Equilibrium Matching model $f$ without having to introduce new parameters.

**Dot Product.** The first approach uses the dot product between the input $x_\gamma$ and the output $f(x_\gamma)$, defined as $g(x_\gamma) = x_\gamma \cdot f(x_\gamma)$. The corresponding derivative with respect to $x_\gamma$ is $\nabla g(x_\gamma) = f(x_\gamma) + x_\gamma^T \nabla f(x_\gamma)$.

**Squared $L_2$ Norm.** The second approach uses the squared $L_2$ norm of the output $f(x_\gamma)$ with a factor of one half to simplify coefficients: $g(x_\gamma) = -\frac{1}{2}||f(x_\gamma)||_2^2$. The corresponding derivative is $\nabla g(x_\gamma) = -f(x_\gamma)\nabla f(x_\gamma)$.

## 3.3 SAMPLING

Because of its equilibrium nature, Equilibrium Matching generates samples via optimization on the learned landscape. In contrast to diffusion/flow models that integrate over a fixed time horizon, EqM decouples sample quality from a prescribed trajectory. It formulates the sampling process as a gradient descent procedure and supports adaptive step sizes, optimizers, and compute, offering additional flexibility at inference time.

**Gradient Descent Sampling (GD).** A simple way to sample from an EqM model is to apply vanilla gradient descent. Let $x_k$ denote the sample after $k$ steps, then an update step with step size $\eta$ is:

$$x_{k+1} \leftarrow x_k - \eta \nabla E(x_k), \tag{8}$$

where $\nabla E(x_k)$ is the predicted gradient at $x_k$ and $E$ may be learned implicitly ($\nabla E(x) = f(x)$) or explicitly ($\nabla E(x) = \nabla g(x)$).

**Sampling with Nesterov Accelerated Gradient (NAG-GD).** Building on gradient descent sampling, we can adopt existing optimization techniques in our sampling procedure. As an example, we use Nesterov Accelerated Gradient (Nesterov, 1983), which applies a look-ahead step at each update and evaluates the gradient at that look-ahead point:

$$x_{k+1} \leftarrow x_k - \eta \nabla E(x_k + \mu(x_k - x_{k-1})), \tag{9}$$

where $\mu$ is the look-ahead factor controlling how far to look ahead at each step.

**Sampling with Differential Equations.** EqM also naturally supports integration-based samplers. ODE-based diffusion samplers can be viewed as a special case of our gradient-based method. We discuss this further in Section D.

**Sampling with Adaptive Compute.** Another advantage of gradient-based sampling is that instead of a fixed number of sampling steps, we can allocate adaptive compute per sample by stopping when the gradient norm drops below a certain threshold $g_{\min}$. For step size $\eta$ and threshold $g_{\min}$, we perform sampling steps $x_{k+1} \leftarrow x_k - \eta \nabla E(x_k)$ until $||\nabla E(x_k)||_2 < g_{\min}$. This allows us to adaptively adjust the number of steps for each individual sample and terminate automatically when close to a local minimum.

## 3.4 IMPLEMENTATION

Equilibrium Matching is simple to implement. We provide example pseudocode for training in Algorithm 1 and sampling in Algorithm 2. During training, we first compute an interpolated corrupted sample $x_\gamma$ from noise $\epsilon$, image $x$, and factor $\gamma$, then train the model $f$ to predict the target gradient

**Algorithm 1** Equilibrium Matching Training

The loss function takes as input model $f$, noise $\epsilon$ (eps), image $x$, and an interpolation factor $\gamma$ (g), and returns the EqM loss.

```
def training_loss(f, eps, x, g):
  xg = (1-g)*eps + g*x
  target = (eps-x)*c(g)
  loss = (f(xg) - target)**2
  return loss
```

**Algorithm 2** Equilibrium Matching Sampling

The sampling function takes as input pretrained model $f$, initialization $st$, step size $\eta$, and total steps $N$, and returns the generated sample.

```
def generate(f, st, eta, N):
  xn = st
  for i in range(N):
    xn = xn - eta*f(xn)
  return xn
```

$(\epsilon - x)c(\gamma)$ by minimizing a mean squared error objective. At inference time, Equilibrium Matching uses the predicted gradients to iteratively optimize via gradient descent.

We adopt a transformer-based backbone from Ma et al. (2024) to implement our Equilibrium Matching model. We use the exact model implementation from Ma et al. (2024) to ensure that no architectural differences influence the results. To remove conditioning on $t$ from this backbone, we set the input $t$ to 0. For further details on model configurations, see Section A.

## 4 ANALYSIS

We provide mathematical justifications for Equilibrium Matching. We show that under common assumptions (Chen et al., 2022; 2023; Lee et al., 2023), Equilibrium Matching learns ground-truth samples as local minima and converges to these minima with a bounded convergence rate. Our analysis is based on a finite discrete dataset and serves as a theoretical approximation of EqM's dynamics.

**Statement 1** (Learned Gradient at Ground-Truth Samples). *Let $f$ be an Equilibrium Matching model with $c(1) = 0$, and let $x^{(i)}$ be a ground-truth sample in $\mathbb{R}^d$. Assume perfect training, i.e., $f$ exactly minimizes the training objective. Then, in high-dimensional settings, we have:*

$$\|f(x^{(i)})\|_2 \approx 0.$$

*where $x^{(i)}$ is an arbitrary sample from the training dataset. In other words, Equilibrium Matching assigns ground-truth images with approximately 0 gradient. (Derivation in Section C.1)*

**Statement 2** (Property of Local Minima). *Let $f$ be an Equilibrium Matching model with $c(1) = 0$, and let $\hat{x}$ be an arbitrary local minimum where $f(\hat{x}) = 0$. Assume perfect training, i.e., $f$ exactly minimizes the training objective. Then, in high-dimensional settings, we have:*

$$P(\hat{x} \in \mathcal{X}) \approx 1.$$

*where $\mathcal{X}$ is the ground-truth dataset. In other words, all local minima are approximately samples from the ground-truth dataset. (Derivation in Section C.2)*

Combining Statement 1 and Statement 2, Equilibrium Matching learns ground-truth images as local minima during training. Next, we show that sampling on an Equilibrium Matching model with gradient descent converges at a rate of $O(\frac{1}{N})$, where $N$ is the total number of sampling steps.

**Statement 3** (Convergence of Gradient-Based Sampling). *Let $f$ be an Equilibrium Matching model with corresponding energy function $E$ such that $\nabla E(x) = f(x)$. Suppose $E$ is $L$-smooth and bounded below by $E(x) \geq E_{inf}$. Then, gradient descent with step size $\eta \in [0, \frac{1}{L}]$ satisfies:*

$$\min_{0 \leq k < K} \|f(x_k)\|^2 \leq \frac{2(E(x_0) - E_{inf})}{\eta K},$$

*where $x_k$ is the iterate after $k$ steps and $K$ is the total number of optimization steps performed. (Derivation in Section C.3)*

We have demonstrated theoretically that under the given assumptions, Equilibrium Matching produces samples close to ground truths. Next, we empirically validate our method.

## 5 EXPERIMENTS

We validate the practical performance of Equilibrium Matching from four major perspectives. First, we demonstrate the advantages in generation quality through a series of experiments on ImageNet

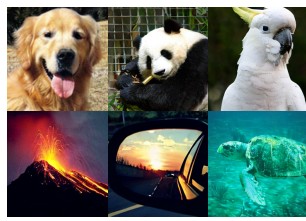

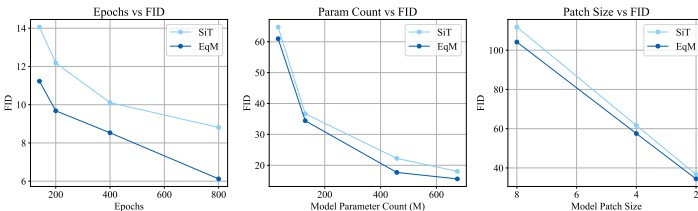

Figure 2: **Curated Samples.** We present curated samples generated by our EqM-XL/2 model.

Figure 3: **Scalability of Equilibrium Matching.** EqM scales across training epochs (left), parameter count (middle), and patch size (right), and outperforms Flow Matching at all tested scales by a significant margin.

| model | method | FID |
|---|---|---|
| StyleGAN-XL | GAN | 2.30 |
| VDM++ | Diffusion | 2.12 |
| DiT-XL/2 | Diffusion | 2.27 |
| SiT-XL/2 | FM | 2.06 |
| **EqM-XL/2** | **EqM** | **1.90** |

| model | sampler | $\eta$ | $\mu$ | FID |
|---|---|---|---|---|
| SiT-XL/2 | Euler (ODE) | 0.0040 | - | 2.10 |
| SiT-XL/2 | Heun (SDE) | 0.0040 | - | 2.06 |
| EqM-XL/2 | Euler (ODE) | 0.0017 | - | 1.93 |
| EqM-XL/2 | GD | 0.0017 | - | 1.93 |
| **EqM-XL/2** | NAG-GD | 0.0017 | 0.3 | **1.90** |

Table 1: **Class-Conditional ImageNet 256×256 Generation.** EqM-XL/2 achieves a 1.90 FID, surpassing other tested methods.

Table 2: **Sampler Comparison.** EqM exceeds Flow Matching in performance (measured by FID) using both integration-based ODE sampler and gradient-based samplers.

(Deng et al., 2009). Then, we examine the properties and performance of our gradient-based sampling method. Next, we illustrate the effectiveness of our gradient landscape via ablation studies. Finally, we show unique properties of Equilibrium Matching that are not inherently supported by diffusion/flow methods. For details of the training and sampling settings used in our experiments, see Section A.

## 5.1 IMAGE GENERATION

**ImageNet Results.** We report performance on class-conditional ImageNet (Deng et al., 2009) 256×256 image generation. We compare Equilibrium Matching with prior generative methods, including StyleGAN (Sauer et al., 2022), VDM++ (Kingma and Gao, 2023), DiT (Peebles and Xie, 2023), and SiT (Ma et al., 2024). Results are shown in Table 1. Equilibrium Matching achieves an FID of 1.90, outperforming both diffusion and flow counterparts by a significant margin across all tested models. We also plot the evolution of FID over time while training our EqM-XL/2 model and compare it with the training FID curve of SiT-XL/2 in Fig. 3. Equilibrium Matching consistently improves over the Flow Matching baseline throughout training, further demonstrating that it produces higher-quality samples than existing generative methods.

**Visualizations.** We present curated samples from our EqM-XL/2 model in Fig. 2 and visualizations of the sampling process in Fig. 4. For both EqM and FM, we use XL/2 models trained for 1400 epochs to produce the visualizations. In Fig. 4, we observe that EqM converges much faster than its FM counterpart at inference. In Fig. 5, we show the top-3 nearest neighbors (measured by mean squared distance) in the training set for EqM-generated samples. The nearest neighbors differ from the generated samples, indicating that EqM does not only memorize the training data and can generalize to unseen samples at inference time.

**Scalability.** To assess scalability, we vary training length, model size, and patch size. We report the evolution of FID over training using the XL/2 training curves. For model-size scaling, we fix the patch size to 2 and the number of training epochs to 80, and evaluate all four model sizes: S, B, L, and XL. For patch-size scaling, we fix the model size to B and training to 80 epochs, and evaluate patch sizes 8, 4, and 2. As shown in Fig. 3, Equilibrium Matching scales well along all axes and consistently outperforms Flow Matching under all tested configurations. These results suggest that Equilibrium Matching has strong scaling potential and is a promising alternative to Flow Matching.

## 5.2 INFERENCE-TIME EXPERIMENTS

**Different Gradient Samplers.** We evaluate our proposed gradient-based samplers on ImageNet generation using the EqM-B/2 model. For NAG-GD, we use $\mu = 0.35$, which we found empirically

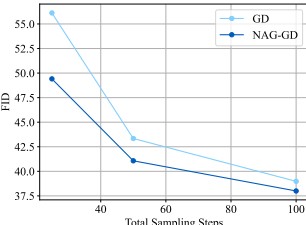
FM

EqM

Figure 4: **Sampling Process Visualization.** We present intermediate samples from XL/2 models using the same 0.004 step size. EqM (bottom) produces realistic images much earlier than FM (top).

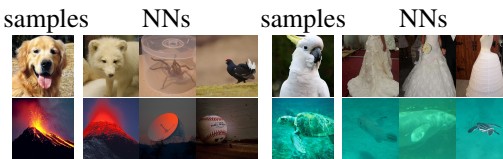
samples    NNs    samples    NNs

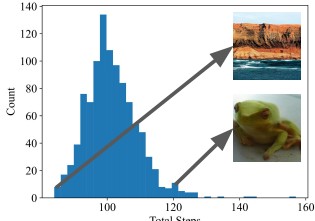

Figure 5: **Nearest Neighbors (NNs) of EqM Samples in the Training Set.** EqM produces samples that are not in the training set, suggesting that it generalizes and does not only memorize the training images.

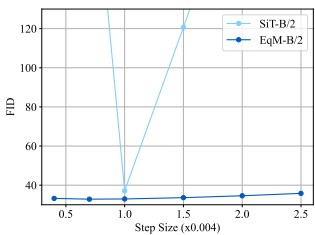

Figure 6: **Sampling with Nesterov Accelerated Gradient.** NAG-GD achieves better sample quality than GD, with the gap being more significant when using fewer steps.

Figure 7: **Different Sampling Step Sizes.** EqM is robust to a wide range of step sizes, whereas Flow Matching only functions properly at one specific step size.

Figure 8: **Total Steps Under Adaptive Compute.** EqM assigns different numbers of steps for different samples, adaptively adjusting compute at inference time.

to work best. As shown in Fig. 6, NAG-GD yields significantly improved FID across all tested step counts. Moreover, the quality gap increases as the total number of steps decreases. This aligns with our intuition: with fewer steps, gradient descent requires more assistance to reach a desirable local minimum, making NAG more effective. In Table 2, we also compare traditional integration samplers with the proposed gradient descent samplers in Equilibrium Matching. Euler ODE sampler, plain gradient descent, and NAG-GD all exceed the Flow Matching baseline by a large margin. These results show that the NAG technique, commonly used in optimization, is also effective during sampling in Equilibrium Matching, providing further evidence that our approach enables new opportunities at inference time. We also provide results using the Adam optimizer (Kingma and Ba, 2014) in Table 12, demonstrating the potential for second-order optimizers in EqM sampling.

**Flexible Step Size.** Viewing sampling through an optimization perspective implies that the step size can be adjusted freely. We evaluate this claim by varying the sampling step size on EqM-B/2. For comparison, we also report the performance of the FM baseline, where we use $\eta$ to replace the ODE update length at each step. We use a total of $N = 250$ sampling steps. From Fig. 7, we observe that our EqM model's generation quality remains high and exceeds the Flow Matching baseline across all tested step sizes. By contrast, Flow Matching requires a specific step size of $\eta = 0.004 = \frac{1}{N}$ to function properly, and small fluctuations in step size lead to significantly worse performance. This suggests that EqM constructs a fundamentally different landscape than FM, which enables new sampling schemes not supported by FM models.

**Adaptive Compute.** We test our adaptive compute sampling using a gradient norm threshold of 10 on EqM-B/2 model. We observe that EqM is able to generate reasonably good samples by adaptive compute, achieving a reasonable FID of 33.79 (32.85 without adaptive compute) using the

| model | FM | FM | EqM | EqM |
|---|---|---|---|---|
| sampler | Euler ODE | Heun SDE | GD | NAG-GD |
| sampling steps | 250 | 250 | 250 | 250 |
| wall-clock time | 70s | 272s | 59s | 55s |
| FID | 2.10 | 2.06 | 1.93 | 1.90 |

Table 3: **Sampling Time Comparison.** We compare the wall-clock time required for different sampling methods in FM and EqM. EqM samples faster than FM while achieving better sample quality.

EqM-B/2 model. We present the distribution of total sampling steps for 1024 samples in Fig. 8, which suggests that EqM assigns different inference-time compute for different samples and manages to lower the total compute to 40% of the original compute (original sampling uses fixed 250 steps). Our results offer promising evidence that EqM can enable new inference-time improvements.

**Sampling Time.** For a more quantitative analysis on sampling compute, we report the wall-clock time required for each sampling approach in Table 3. We use the XL/2 model to generate 50000

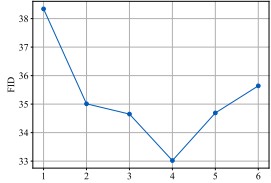

Figure 9: **Different Gradient Multipliers.** $\lambda = 4$ performs the best (32.85).

Table 5: **Noise Conditioning Ablation.** Our truncated decay $c_{\text{trunc}}$ improves generation only when not conditioned on noise level, matching our hypothesis.

| noise conditioning | yes | yes | no | no |
|---|---|---|---|---|
| $c(\gamma)$ | const | trunc | const | trunc |
| FID | 36.68 | 41.89 | 40.81 | **32.85** |

Table 6: **EqM-E Variants.** The dot product variant outperforms $L_2$ norm.

| energy model | FID |
|---|---|
| none | 57.54 |
| dot product | 73.40 |
| $L_2$ norm | 75.53 |

samples using identical batch size for all methods, and we report the averaged time per batch of 64 samples on a single H100 GPU. We find that EqM not only achieves better FID but also samples faster than typical differential-equation-based samplers.

### 5.3 ABLATION STUDY

**Hyperparameter Choices.** To determine the best target landscape, we search over choices and hyperparameters for $c(\gamma)$. We use our EqM-B/2 model and 80 epochs of training for all hyperparameter experiments. We tune the step size

Table 4: **Different Target Gradient Fields.** Several settings exceed the noise unconditional Flow Matching baseline in performance. Best performance achieved with the truncated decay $c_{\text{trunc}}$ and hyperparameter $a = 0.8$.

| $c(\gamma)$ | constant | linear | truncated | | | piecewise | |
|---|---|---|---|---|---|---|---|
| $a$ | - | - | 0.5 | 0.8 | 0.9 | 0.8 | 0.8 |
| $b$ | - | - | - | - | - | 0.8 | 1.4 |
| FID | 40.81 | 50.47 | 38.98 | **38.34** | 41.22 | 38.84 | 38.75 |

of GD sampler and report the best result for each setting in Table 4. The best-performing gradient landscape is truncated decay with $a = 0.8$. Our results suggest that it is helpful to keep a constant target gradient at the start of the trajectory before decaying to 0. We then sweep the gradient multiplier $\lambda$ on this best-performing gradient field. As shown in Fig. 9, a multiplier of 4 significantly improves FID. We use this setting ($c_{\text{trunc}}$ with $a = 0.8$ and $\lambda = 4$) as default in our experiments.

**Noise Conditioning.** We compare our new target gradient with the baseline under both noise-conditional and noise-unconditional settings in Table 5. We adopt the 80 epochs B/2 model for both the FM baseline and EqM. Our target gradient improves performance only in the noise-unconditional case, which aligns with our expectation that an energy landscape with zero gradient at real samples is more favorable under equilibrium dynamics.

**Energy Formulations.** We evaluate two EqM-E variants, the dot product and the $L_2$ norm, on the EqM-B/4 model. We train the dot product EqM-E model from scratch for 80 epochs, and we train the $L_2$ norm model by first initializing from a pretrained EqM model (10 epochs) and then continuing training for 70 epochs. We adopt this scheme due to stability concerns, as the $L_2$ norm variant is sensitive to initialization. Results are presented in Table 6. We also report the evolution of energy value using the dot product variant in Fig. 12. We find that both formulations degrade performance, which we attribute to optimization difficulties stemming from second-order differentiation. Among the two, the $L_2$ norm variant performs worse. Since it also requires careful initialization to train stably, we conclude that the $L_2$ norm variant is harder to optimize than the dot product variant of Equilibrium Matching. We attribute the overall degeneration in performance to the inherent difficulty of optimizing a single energy value, which aligns with prior difficulties on training EBMs. Consequently, we recommend using the dot product variant for explicit energy.

### 5.4 PROPERTIES OF EQUILIBRIUM MATCHING

In this subsection, we investigate unique properties of Equilibrium Matching that are not supported by traditional diffusion/flow models.

**Partially Noised Image Denoising.** By learning an equilibrium dynamic, Equilibrium Matching can directly start with and denoise a partially noised image. Existing diffusion/flow models require an explicit noise level as input to process partially noised images, but our EqM model does not have such a limitation. We evaluate EqM's generation quality from partially noised inputs using noised samples from the ImageNet validation set. As shown in Fig. 10, Equilibrium Matching behaves very differently from traditional Flow Matching. EqM-B/4's FID improves significantly when fed less

noisy samples, whereas the Flow-based SiT-B/4 cannot handle partially noised images as raw input and its generation quality drops quickly when not starting from pure noise. These results further support that Equilibrium Matching enables capabilities that traditional methods cannot naturally offer.

**Out-of-Distribution Detection.** Another unique property of the EqM model is its inherent ability to perform out-of-distribution (OOD) detection using energy value. In-distribution (ID) samples typically have lower energies than OOD samples. To this end, we use our dot product variant of the EqM-E-B/4 model and perform OOD detection with CIFAR-10 as ID. We use PixelCNN++ (Salimans et al., 2017), GLOW (Kingma and Dhariwal, 2018), and IGEBM (Du and Mordatch, 2019) as baselines and adopt the numbers reported by Yoon et al. (2023). We report the area under the ROC curve (AUROC) in Table 8. Compared with these baselines, Equilibrium Matching provides reasonable OOD detection across all tested datasets and achieves the best overall performance, suggesting that Equilibrium Matching indeed learns a valid energy landscape.

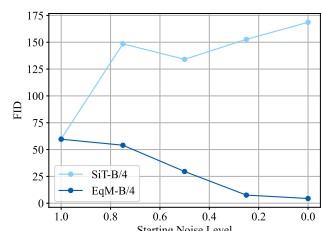

Figure 10: **Partially Noised Image Generation.** EqM's generation quality improves with less noisy inputs while FM's quality drops.

**Composition.** EqM also naturally supports the composition of multiple models by adding energy landscapes together (corresponding to adding the gradients of each model). We test composition by combining models conditioned on different ImageNet class labels. We use our EqM-XL/2 model with GD sampler and add two conditional gradients together as the update gradient at each sampling step. In Fig. 11, we present the generation results using panda and valley (top left), car mirror and volcano (top right), ice cream and chocolate syrup (bottom left), and broccoli and cauliflower (bottom right). These results demonstrate that EqM is easily composable by optimizing the summed gradient. This is similar to the composability of EBMs (Du et al., 2020a), while the composition of diffusion is significantly more complex to accurately implement (Du et al., 2023).

## 6 RELATED WORK

**Diffusion Models and Flow Matching.** Diffusion models (Sohl-Dickstein et al., 2015; Ho et al., 2020; Song et al.; Nichol and Dhariwal, 2021; Dhariwal and Nichol, 2021; Karras et al., 2022) generate images from pure noise through a series of noising and denoising steps that are conditioned on noise level. The sampling process of diffusion models is often formulated as solving a differential equation by integrating the model's predicted velocity over noise level (Ho et al., 2020; Song et al., 2020; Lu et al., 2022; Karras et al., 2022). Flow Matching (Lipman et al.; Liu et al.; Albergo and Vanden-Eijnden, 2023) is a more recent generative method that adopts a linear interpolation between noise and real images, which eliminates the need for complex noise scheduling.

**Energy-Based Models.** Energy-based models (EBMs) (Hinton, 2002; LeCun et al., 2006; Xie et al., 2016; Du and Mordatch, 2019; Du et al., 2020b; Nijkamp et al., 2020; Gao et al., 2020) learn an energy landscape that defines the unnormalized log-density of data distribution. EBMs are versatile in different modalities and tasks (e.g., OOD detection) thanks to the equilibrium energy landscape (Du and Mordatch, 2019; Grathwohl et al., 2019). However, EBMs suffer from training instabilities (Carreira-Perpinan and Hinton, 2005; Song and Ou, 2018; Gutmann and Hyvärinen, 2010) and are hard to scale (Du et al., 2020b).

**Existing Efforts.** Prior efforts on improving generative modeling have attempted to merge diffusion and energy training. In order to make diffusion learn equilibrium dynamics, Sun et al. (2025) attempt to directly remove noise conditioning from diffusion models, but this leads to worse generation quality. Energy Matching (Balcerak et al., 2025) adopts a two-stage training strategy where the model is first trained using a flow objective and then trained with Langevin-based dynamics like EBM near the data manifold. However, Energy Matching is outperformed by Flow Matching on large-scale experiments like ImageNet. Our work is different from Energy Matching in that we train with a single objective that unifies the dynamics near and away from data. Contrary to Energy Matching, EqM offers promising scalability, optimization-based sampling, and additional flexibility.

To clarify how EqM differs from prior work on implicit energy landscapes and hybrid flow/EBM methods, we summarize the key distinctions in Table 7. As shown, EqM removes all time/noise

| model | SVHN | Textures | constant | avg. |
|---|---|---|---|---|
| Pixel-CNN++ | 0.32 | 0.33 | 0.71 | 0.45 |
| GLOW | 0.24 | 0.27 | - | 0.26 |
| IGEBM | **0.63** | 0.48 | 0.39 | 0.50 |
| **EqM** | 0.55 | **0.49** | **1.00** | **0.68** |

Table 8: **OOD Detection.** EqM achieves reasonable AUROC under all tested OOD datasets and has the best average result among all tested models.

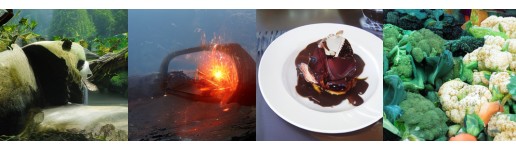

Figure 11: **Image Composition.** We present compositional samples from EqM-XL/2 each using two ImageNet labels: panda and valley (leftmost), car mirror and volcano (second left), ice cream and chocolate syrup (second right), and broccoli and cauliflower (rightmost).

conditioning and directly learns a stationary equilibrium gradient field, whereas Energy Matching (Balcerak et al., 2025), VAPO (Yue et al., 2025), and Action Matching (Neklyudov et al., 2023) all operate in explicitly time- or noise-conditioned settings and optimize time-dependent trajectories or actions rather than a single equilibrium landscape. Due to difficulties training EBMs on ImageNet, we provide a quantitative comparison between EqM and other EBM methods on CIFAR-10 in Table 10.

| method | time conditioning | objective |
|---|---|---|
| EqM (ours) | no | a single equilibrium energy landscape $E(x)$ |
| Energy Matching | yes | energies or scores of a reference diffusion/EBM |
| VAPO | yes | variational objectives over time-dependent trajectories |
| Action Matching | yes | actions of a reference flow |

Table 7: **Conceptual comparison between EqM and related hybrid flow/EBM methods.**

# 7 CONCLUSION

We propose Equilibrium Matching, a generative model that learns equilibrium dynamics in a simple and effective way. Equilibrium Matching combines the advantages of energy-based and flow-based models without compromising performance. Our method is easy to train, achieves strong generation quality, and provides an interpretable energy landscape that supports a wide range of sampling methods. We hope that the equilibrium dynamics learned by Equilibrium Matching will inspire more effective and scalable inference algorithms in the future.

## ETHICS AND REPRODUCIBILITY STATEMENTS

We strictly adhere to the ICLR Code of Ethics. We use open-sourced datasets and models for all of our experiments, and we strictly follow existing evaluation protocols.

For reproducibility, we include our code in the supplementary materials. We also include a detailed README file that describes how to reproduce our results.

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

# A  EXPERIMENTAL SETTING

## A.1  TRAINING SETTING

We present the training setting of our Equilibrium Matching model in Table 9.

## A.2  INFERENCE SETTING

We present the majority of our sampler settings in Table 9. More specifically, in Section 5.1, we adopt the SiT results reported by Ma et al. (2024) and Wang and He (2025), and use our own NAG-GD sampler for EqM results. In Section 5.2, Section 5.3, and Section 5.4, we adopt the Euler sampler for SiT experiments and the vanilla GD sampler for EqM experiments.

# B  ADDITIONAL EXPERIMENTS

## B.1  CIFAR-10 EXPERIMENTS

We also evaluate our approach on non-transformer architectures in the CIFAR-10 dataset (Krizhevsky, 2009), where the commonly used network architecture is U-Net (Ronneberger et al., 2015). The experiments are based on the publicly available code of Flow Matching (Lipman et al., 2024). We use the same hyper-parameters as the original codebase, with the only difference being that we do not use EDM scheduling or skewed timesteps, which are tricks designed specifically for flow matching training (Karras et al., 2022). Table 10 presents the FID results. We used the truncated decay function with $a = 0.4, \lambda = 2.0$ for our EqM model. Equilibrium Matching outperforms Flow Matching baseline, similar to our observations on SiT. This demonstrates that EqM is a general generative modeling approach applicable across different datasets and model backbones.

Since it has been hard to directly train EBMs on ImageNet and most existing literature only report CIFAR-10 numbers, we compare other related methods' performance on CIFAR-10 in Table 10, including IGEBM (Du and Mordatch, 2019; Yang et al., 2023), Energy-based U-net (Salimans and Ho, 2021), and Energy Matching (Balcerak et al., 2025). EqM outperforms all listed methods, making it a promising approach for generative modeling.

## B.2  OTHER METRICS

We report Equilibrium Matching's performance on other evaluation metrics including sFID and Inception Score (IS) in Table 11. Equilibrium Matching also achieves relatively good sFID and IS compared against other generative methods.

## B.3  ENERGY EVOLUTION

We include an energy evolution plot over the sampling process for our EqM-E model in Fig. 12. We used the B/4 model with dot product parameterization and report the average energy curve over 64 samples. We see that the energy value consistently and smoothly decreases over the course of sampling, which is consistent with the expected behavior of an EBM.

## B.4  SECOND-ORDER OPTIMIZER

We also attempt to directly adopt existing second-order optimizers for EqM sampling. In particular, we use Adam without momentum ($\beta_1 = \beta_2 = 0$). We find that Adam performs decently well on EqM-B/2, as shown in Table 12. This opens exciting opportunities for future research by demonstrating that reasonable samples can still be produced when the gradients are normalized.

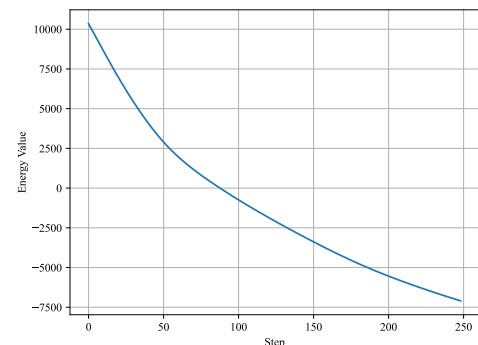

Figure 12: **Energy Evolution During Sampling.** We report the energy curve during sampling on EqM-E. The energy value decreases smoothly and consistently, matching our expectation.

## C DERIVATIONS

### C.1 LEARNED GRADIENT AT GROUND-TRUTH SAMPLES

**Statement 1** (Learned Gradient at Ground-Truth Samples). *Let $f$ be an Equilibrium Matching model with $c(1) = 0$, and let $x^{(i)}$ be a ground-truth sample in $\mathbb{R}^d$. Assume perfect training, i.e., $f$ exactly minimizes the training objective. Then, in high-dimensional settings, we have:*

$$\|f(x^{(i)})\|_2 \approx 0.$$

*where $x^{(i)}$ is an arbitrary sample from the training dataset. In other words, Equilibrium Matching assigns ground-truth images with approximately 0 gradient.*

*Derivation of Statement 1.* Under perfect training, the squared-error objective

$$\mathcal{L} = \mathbb{E}_{x,\epsilon,\gamma}\big\|f(x_\gamma) - (\epsilon - x)\,c(\gamma)\big\|^2$$

is minimized when

$$f(x_\gamma) \;=\; \mathbb{E}\big[(\epsilon - x)\,c(\gamma) \;\big|\; x_\gamma\big]. \tag{1}$$

We use the forward noising model

$$x_\gamma \;=\; \gamma\,x \;+\; (1-\gamma)\,\epsilon,$$

where $x \in \mathcal{X}$ is one of finitely many training points and $\epsilon \sim \mathcal{N}(0, I_d)$. For any fixed $\gamma$ and $x$, the conditional density of $x_\gamma$ given $t$,

$$p(x_\gamma \mid \gamma) \;=\; \mathcal{N}\big(\gamma\,x,\,(1-\gamma)^2 I_d\big),$$

is a continuous Gaussian on $\mathbb{R}^d$.

At $\gamma = 1$, $x_\gamma$ equals exactly $x$ with probability one (a Dirac mass on each $x \in \mathcal{X}$).

Since $\mathcal{X}$ is a finite discrete set, in the limit $d \to \infty$ the Gaussian density at any exact training point $x^{(i)}$ for $\gamma < 1$ vanishes exponentially in $d$, whereas the mass at $\gamma = 1$ remains. Hence

$$P\big(\gamma = 1 \mid x_\gamma = x^{(i)}\big) \;\longrightarrow\; 1 \quad (d \to \infty). \tag{2}$$

Plugging equation 2 into equation 1, we get

$$f\big(x^{(i)}\big) \;=\; \mathbb{E}\big[(\epsilon - x)\,c(\gamma) \;\big|\; x_\gamma = x^{(i)}\big] \;\approx\; (\epsilon - x^{(i)})\,c(1) \;=\; 0,$$

since $c(1) = 0$. Therefore $\|f(x^{(i)})\|_2 \approx 0$, as claimed. $\qquad\square$

| model | S/2 | B/2 | L/2 | XL/2 |
|---|---|---|---|---|
| **model configurations** | | | | |
| params (M) | 33 | 130 | 458 | 675 |
| depth | 12 | 12 | 24 | 28 |
| hidden dim | 384 | 768 | 1024 | 1152 |
| patch size | 2 | 2 | 2 | 2 |
| heads | 6 | 12 | 16 | 16 |
| **ImageNet training configurations** | | | | |
| epochs | 80 | 80 | 80 | 80 - 1400 |
| batch size | | 256 | | |
| optimizer | | AdamW | | |
| optimizer $\beta_1$ | | 0.9 | | |
| optimizer $\beta_2$ | | 0.999 | | |
| weight decay | | 0.0 | | |
| learning rate (lr) | | $1 \times 10^{-4}$ | | |
| lr schedule | | constant | | |
| lr warmup | | none | | |
| **gradient field hparams** | | | | |
| choice of $c(\gamma)$ | | truncated decay | | |
| $c(\gamma)$ multiplier $\lambda$ | | 4.0 | | |
| $a$ | | 0.8 | | |
| $b$ | | - | | |
| **integration sampler configurations** | | | | |
| sampler | dopri5 | dopri5 | dopri5 | Euler |
| NFE | | 250 | | |
| $\eta$ | 0.004 | 0.004 | 0.004 | 0.0017 |
| **gradient sampler configurations** | | | | |
| sampler | | GD/NAG-GD | | |
| steps | | 250 | | |
| $\eta$ | 0.003 | 0.003 | 0.003 | 0.0017 |
| $\mu$ | 0.35 | 0.35 | 0.35 | 0.3 |

Table 9: **Equilibrium Matching Configurations.**

## C.2 PROPERTY OF LOCAL MINIMA

**Statement 2** (Property of Local Minima). *Let $f$ be an Equilibrium Matching model with $c(1) = 0$, and let $\hat{x}$ be an arbitrary local minimum where $f(\hat{x}) = 0$. Assume perfect training, i.e., $f$ exactly minimizes the training objective. Then, in high-dimensional settings, we have:*

$$P(\hat{x} \in \mathcal{X}) \approx 1.$$

*where $\mathcal{X}$ is the ground-truth dataset. In other words, all local minima are approximately samples from the ground-truth dataset.*

*Derivation of Statement 2.* By the same argument as before, perfect training implies

$$0 = f(\hat{x}) = \mathbb{E}\big[(\epsilon - x)\, c(\gamma) \mid x_\gamma = \hat{x}\big]. \tag{1}$$

Since $c(\gamma) \geq 0$ for all $\gamma$ and $c(1) = 0$ while $c(\gamma) > 0$ for $\gamma < 1$, the only way the vector-valued expectation in equation 1 can vanish in high dimension is if the posterior mass concentrates at $\gamma = 1$.

We can argue exactly as in equation 2: for any $\hat{x}$ that equals some $x^{(i)} \in \mathcal{X}$, we have

$$P(\gamma = 1 \mid x_\gamma = \hat{x}) \longrightarrow 1,$$

and for $\gamma < 1$ the density is exponentially small in $d$. Since at $\gamma = 1$, all $x_\gamma$ are in $\mathcal{X}$,

$$P(\hat{x} \in \mathcal{X}) \approx 1.$$

establishing the claim. $\qquad\qquad\qquad\qquad\qquad\qquad\qquad\qquad\qquad\qquad\qquad\qquad\qquad\qquad\square$

| model | FM | IGEBM | Energy-based U-net | Energy Matching | EqM |
|-------|-----|-------|-------------------|-----------------|-----|
| FID | 3.70 | 37.9 | 6.8 | 3.34 | 3.32 |

Table 10: **Image Generation on CIFAR-10.** We report FID on CIFAR-10 class-conditional image generation with U-Net. EqM improves upon Flow Matching and energy-based methods on CIFAR-10 generation.

| model | method | FID ↓ | sFID ↓ | IS ↑ |
|-------|--------|-------|--------|------|
| StyleGAN-XL | GAN | 2.30 | **4.02** | 265.1 |
| VDM++ | Diffusion | 2.12 | - | 267.7 |
| DiT-XL/2 | Diffusion | 2.27 | 4.60 | **278.2** |
| SiT-XL/2 | FM | 2.06 | 4.49 | 277.5 |
| **EqM-XL/2** | **EqM** | **1.90** | 4.54 | 275.7 |

Table 11: **Class-conditional Generation on ImageNet 256×256 with Additional Metrics.** Equilibrium Matching achieves the best FID and relatively good sFID and IS.

## C.3 CONVERGENCE OF GRADIENT-BASED SAMPLING

**Statement 3** (Convergence of Gradient-Based Sampling). *Let $f$ be an Equilibrium Matching model with corresponding energy function $E$ such that $\nabla E(x) = f(x)$. Suppose $E$ is L-smooth and bounded below by $E(x) \geq E_{inf}$. Then, gradient descent with step size $\eta \in \left[0, \frac{1}{L}\right]$ satisfies:*

$$\min_{0 \leq k < K} \|f(x_k)\|^2 \leq \frac{2\big(E(x_0) - E_{inf}\big)}{\eta\, K},$$

*where $x_k$ is the iterate after $k$ steps and $K$ is the total number of optimization steps performed.*

*Derivation of Statement 3.* By $L$-smoothness of $E$, for any $x, y \in \mathbb{R}^d$,

$$E(y) \leq E(x) + \nabla E(x)^\top (y - x) + \frac{L}{2}\|y - x\|^2.$$

Take $y = x_{k+1} = x_k - \eta\, f(x_k)$ and recall $\nabla E(x_k) = -f(x_k)$. Then

$$E(x_{k+1}) \leq E(x_k) - \eta\, f(x_k)^\top f(x_k) + \frac{L}{2}\eta^2 \|f(x_k)\|^2$$
$$= E(x_k) - \eta\Big(1 - \frac{L\eta}{2}\Big)\|f(x_k)\|^2 \ \geq\ E(x_k) - \frac{\eta}{2}\|f(x_k)\|^2,$$

where the last inequality holds because $\eta \leq 1/L \implies 1 - \frac{L\eta}{2} \geq \frac{1}{2}$.

Summing from $k = 0$ to $k = K - 1$ gives

$$E(x_K) - E(x_0) \ \leq\ -\frac{\eta}{2}\sum_{k=0}^{K-1} \|f(x_k)\|^2 \ \leq\ -\frac{\eta}{2}\, K \min_{0 \leq k < K} \|f(x_k)\|^2.$$

Since $E(x_K) \geq E_{\inf}$, rearrange to obtain

$$\min_{0 \leq k < K} \|f(x_k)\|^2 \ \leq\ \frac{2\big(E(x_0) - E_{\inf}\big)}{\eta\, K},$$

as required. □

## D RELATION BETWEEN INTEGRATION-BASED AND OPTIMIZATION-BASED SAMPLING

### D.1 ODE SAMPLING AS A SPECIAL CASE OF OPTIMIZATION

Consider the ODE

$$\dot{x} = v(x),$$

| sampler | GD | NAG-GD | Adam |
|---------|-----|--------|------|
| FID | 32.85 | 32.97 | 36.35 |

Table 12: **EqM Sampling wth Adam.** We find that Adam without momentum achieves decent generation quality on EqM-B/2.

and its explicit (forward) Euler discretization with a uniform time grid on $[0, 1]$:

$$x_{k+1} = x_k + h\, v(x_k), \qquad h = \tfrac{1}{N},\; k = 0, \ldots, N-1.$$

When the velocity field is conservative with potential $E$, i.e., $v(x) = -\nabla E(x)$, the Euler step becomes:

$$x_{k+1} = x_k - h\, \nabla E(x_k),$$

which is exactly a gradient-descent update with step size $\eta = h = \tfrac{1}{N}$. Hence, with $N$ steps on a unit time horizon, the gradient-descent sampler with $\eta = \tfrac{1}{N}$ coincides with the explicit Euler ODE sampler.

### D.2 GENERAL INTEGRATION SAMPLERS

The equivalence above suggests a broader correspondence: integration-based samplers can be interpreted as optimization-based methods by viewing the velocity as a descent direction. In particular, when $v(x) = -\nabla E(x)$, any time integrator induces an optimization update rule.

Practically, ODE samplers in generative modeling are often implemented on a uniform grid with step size $h = \tfrac{1}{N}$. Under the optimization view, we are not bound to this constraint. We can adopt adaptive step sizes while retaining the same underlying direction field. This perspective enables a direct adaptation of existing integration-based samplers within Equilibrium Matching.

## E  LLM USAGE

LLMs are only used to correct grammar mistakes and are not used for research ideation or writing.

