# OpenReview forum: "Equilibrium Matching: Generative Modeling with Implicit Energy-Based Models"
_ICLR.cc/2026/Conference — Submitted to ICLR 2026_

### Official Review · Reviewer_Ct1d · 2025-10-30

**Soundness:** 2
**Presentation:** 3
**Contribution:** 2
**Rating:** 4
**Confidence:** 2

**Summary:**

This paper introduces Equilibrium Matching (EqM), a novel generative modeling framework that replaces the non-equilibrium, time-conditional dynamics of diffusion and flow-based models with a single, time-invariant equilibrium gradien. EqM learns the gradient field of an implicit energy landscape by training a noise-unconditional model to match a new target gradient. This target is designed to point from noise to data and, crucially, to vanish at the data manifold, ensuring ground-truth samples are stable local minima. This equilibrium perspective enables flexible, "optimization-based sampling" (e.g., gradient descent, NAG, adaptive compute) at inference time. Empirically, EqM achieves state-of-the-art performance, and demonstrates capabilities absent in standard flow models, such as OOD detection, image composition, and high-fidelity denoising from partially noised inputs.

**Strengths:**

- The shift from a time-conditional velocity field to a single equilibrium gradient field is a simple but powerful conceptual contribution.
- The primary strength of EqM is its strong sample quality. It achieves a 1.90 FID on class-conditional ImageNet 256x256, outperforming the strong SiT (Flow Matching) baseline (2.06 FID) and the DiT (Diffusion) baseline (2.27 FID).

**Weaknesses:**

- The theoretical justification (Statements 1 and 2) that the model's local minima are the ground-truth data samples relies on "perfect training" and "high-dimensional" approximations . While the empirical results are strong, the theory provides an approximation rather than a hard guarantee.
- The paper achieves its SOTA results by applying its new objective to the exact same SiT transformer backbone as its baseline. While this makes for a fair comparison, it also means the paper's contribution is entirely dependent on that specific architecture. Given the CIFAR-10/U-Net failure (Table 8), it is unknown if the EqM objective would provide any benefit to other model families.
- Optimization samplers can be slower per sample than well-tuned ODE/ SDE samplers; the paper claims adaptive savings but does not present thorough wall-clock or FLOP comparisons against accelerated samplers (e.g., DDIM, DDPM with distillation, flow samplers) at matched FID. Practical deployment requires these cost–quality tradeoffs; the paper’s lack of rigorous latency/compute comparisons weakens its practical case.

**Questions:**

- The use of NAG-GD for sampling improves FID (Table 2). Since sampling is just gradient-based optimization, have you explored more advanced optimizers (e.g., Adam, L-BFGS)? Could this optimization-based sampling framework unlock even better sample quality?
- Can you show wall-clock and FLOP comparisons (not just # function evaluations) vs. strong baselines (DDIM, Flow Matching, distilled samplers) at matched FID?

---

> ### Author Response · Authors · 2025-11-20
>
> ### Theoretical Justification
> We thank the reviewer for carefully examining the theoretical aspects. We agree that our theoretical justifications should be viewed as **conceptual approximations**, not as fully realistic guarantees. We adopt standard assumptions used in the generative modeling and optimization literature (e.g., smoothness, well-behaved gradients) [1,2,3], and we acknowledge that it is difficult to rigorously quantify training error and model mismatch in large-scale image generation settings. For this reason, we complement the theoretical analysis with extensive empirical experiments that verify the effectiveness of EqM on real-world datasets. We have modified our Analysis section to reflect these changes.
>
> ### Dependence on SiT Backbone
> We appreciate the reviewer’s comment. We would like to point out that the CIFAR-10 Flow Matching baseline uses an EDM schedule and skewed timesteps [4,5], which are techniques specifically designed to improve CIFAR-10 FM and not part of the original FM algorithm. These techniques are used only in the CIFAR-10 FM baseline but not on ImageNet. In additional experiments, we observe:
>
> |              | FM   | EqM  |
> |--------------|------|------|
> | w/ tricks    | 2.09 | 3.36 |
> | w/o tricks   | 3.70 | 3.32 |
>
> Without these CIFAR-10 FM-specific training tricks, EqM outperforms FM, including on U-Net architectures. This is consistent with our ImageNet results using a transformer backbone and suggests that EqM is **not inherently tied** to SiT; rather, it benefits from training setups aligned with its equilibrium dynamics. We note that there could be equivalent tricks like EDM schedule and skewed timesteps that improves EqM’s performance.
>
> ### Wall-Clock Comparisons
> We thank the reviewer for the suggestion to include wall-clock comparisons. We have added a sampling time comparison in Section 5.2 and Table 3, where we compare EqM and FM in terms of both wall-clock time and generation quality. We report the averaged time per batch of 64 samples on a single H100 GPU. The results are:
>
> | model                   | FM        | FM        | EqM    | EqM     |
> |-------------------------|-----------|-----------|--------|---------|
> | sampler                 | Euler ODE | Heun SDE  | GD     | NAG-GD  |
> | sampling steps          | 250       | 250       | 250    | 250     |
> |wall-clock time|   70s | 272s | 59s | 55s |
> | FID                     | 2.10      | 2.06      | 1.93   | 1.90    |
>
> EqM consistently provides **faster sampling** and **better FID** than the FM baselines.
>
> ### Advanced Optimizers
> We appreciate the suggestion to explore more advanced optimizers. In our additional experiments, we find that Adam without momentum performs reasonably well, achieving a FID of 36.35 on EqM-B/2 (compared to 32.85 FID for simple GD sampling). Although Adam is currently slightly worse than plain gradient descent, we agree that with more careful tuning or second-order variants, advanced optimizers could be a promising direction for future work. We have modified the Appendix to include these results (Table 12).
>
> | sampler                 | GD | Adam |
> |-------------------------|-----------|-----------|
> | FID                     | 32.85      | 36.35       |
>
> [1] Chen et al., “Sampling is as easy as learning the score: theory for diffusion models with minimal data assumptions,” ICLR 2023
> [2] Chen, Lee, Lu, “Improved Analysis of Score-based Generative Modeling: User-Friendly Bounds under Minimal Smoothness Assumptions,” ICML 2023
> [3] Lee, Lu, Tan, “Convergence of Score-based Generative Modeling for General Data Distributions,” ALT 2023
> [4] Lipman, Y., Havasi, M., Holderrieth, P., Shaul, N., Le, M., Karrer, B., Chen, R.T., Lopez-Paz, D., Ben-Hamu, H. and Gat, I., 2024. *Flow matching guide and code.* arXiv preprint arXiv:2412.06264.
> [5] Karras, T., Aittala, M., Aila, T. and Laine, S., 2022. *Elucidating the design space of diffusion-based generative models.* NeurIPS 35, pp. 26565–26577.

---

### Official Review · Reviewer_xen9 · 2025-10-31

**Soundness:** 3
**Presentation:** 3
**Contribution:** 3
**Rating:** 8
**Confidence:** 4

**Summary:**

This paper introduces a novel generative modeling framework — Equilibrium Matching (EqM), which aims to unify Flow Matching (FM) and Energy-Based Models (EBM) from the perspective of equilibrium dynamics. Traditional diffusion or flow models rely on non-equilibrium dynamics conditioned on time, requiring the definition of noise levels and time steps. In contrast, EqM discards this conditional process and directly learns the equilibrium gradient field of the implicit energy landscape.With this setup, EqM enables optimization-driven sampling (such as gradient descent, Nesterov Accelerated Gradient, etc.) during the inference phase, generating high-quality samples without the need for trajectory integration.Overall, this work redefines the structure of generative models through the energy minimization perspective of equilibrium dynamics, providing a new theoretical bridge to connect Flow Matching with Energy-Based Models.

**Strengths:**

1. Novelty：
The key innovations over prior work are: (1) ‌eliminating explicit time/noise conditioning‌ by directly learning equilibrium dynamics, and (2) ‌unifying generative sampling as gradient-based optimization.


2. Mathematically rigorous：
The paper theoretically shows that the Gradient-Based Sampling will converge.


3. SOTA performance：
EqM shows a better FID score over prior work. It demonstrates scalability in high resolution and adaptive inference, while outperforming flow matching models in tasks like ImageNet and OOD detection. This work effectively combines generative modeling and optimization theory, laying the foundation for an energy-based sampling framework with significant future application potential.

**Weaknesses:**

1 Lack of Systematic Comparison with Existing EBM/EM Methods.
Although the authors compare EqM with several energy-based models (e.g., IGEBM, GLOW, PixelCNN++) in Table 6, this evaluation only verifies EqM’s energy discrimination ability. It does not include systematic comparisons in terms of generation quality, energy landscape modeling, or convergence analysis. Moreover, since EqM is theoretically similar to Energy Matching (Balcerak et al., 2025), the absence of direct experimental comparison with that method weakens the validation of EqM’s claimed novelty.

2. Insufficient Analysis of Sampling Mechanisms.
While EqM introduces GD, NAG-GD, and adaptive sampling strategies, the paper presents only qualitative results without a quantitative or theoretical examination of convergence speed, energy evolution, or performance under different energy landscape conditions.

3. Sampling inefficiency. The sampling process still needs an iterative MCMC process, which might be slow in practice. I strongly suggest including sampling time comparison with prior EBM and flow-matching work.

**Questions:**

Q1: The theoretical derivations (Propositions 1–3) rely on the assumption of perfect training, where the model ( f ) fully minimizes its training objective. How would EqM behave when this idealized condition is relaxed — for instance, under optimization noise, incomplete convergence, or limited model capacity?

Q2: Since EqM performs deterministic gradient-based sampling, is sample diversity entirely dependent on the randomness of initialization?

---

> ### Author Response · Authors · 2025-11-20
>
> ### Lack of Systematic Comparison with Existing EBM/EM Methods
> We thank the reviewer for this important suggestion. Since it has been hard to directly train EBMs on ImageNet and most existing literatures only report CIFAR-10 numbers, we have augmented Appendix B.1 (Table 8) with additional CIFAR-10 results comparing EqM against EBMs and Energy Matching. The updated comparison is:
>
> | model              | FM   | IGEBM | Energy-based U-Net | Energy Matching | EqM  |
> |--------------------|------|-------|--------------------|-----------------|------|
> | FID                | 3.70 | 37.9  | 6.8                | 3.34            | 3.32 |
>
> EqM achieves the best FID among all tested models, outperforming both traditional EBMs and Energy Matching.
>
> ### Insufficient Analysis of Sampling Mechanisms
> We appreciate the request for a more detailed discussion of the sampling mechanism. We have added an energy evolution plot in Appendix B.3 Figure 12 (also see https://ibb.co/R4DDMkXm), where we observe consistent decrease in the energy value during sampling. This is consistent with the expected behavior.
>
> ### Sampling Inefficiency
> We thank the reviewer for this valuable suggestion. In Section 5.2 and Table 3, we have added a **sampling time comparison** between EqM and Flow Matching. We report the averaged time per batch of 64 samples on a single H100 GPU. We report both generation quality (FID) and wall-clock time:
>
> | model                   | FM        | FM        | EqM    | EqM     |
> |-------------------------|-----------|-----------|--------|---------|
> | sampler                 | Euler ODE | Heun SDE  | GD     | NAG-GD  |
> | sampling steps          | 250       | 250       | 250    | 250     |
> |wall-clock time|   70s | 272s | 59s | 55s |
> | FID                     | 2.10      | 2.06      | 1.93   | 1.90    |
>
> EqM consistently achieves **better sampling efficiency** (shorter wall-clock time) while also reaching **better FID** than the FM baselines.
>
> ### Relaxed Theoretical Conditions
> We appreciate the question about relaxing theoretical conditions. When training error is present, the local minima of the learned landscape may no longer coincide exactly with the true data points; instead, they can deviate in proportion to the training error. Deriving tight error bounds would require additional assumptions on the noise level, optimization convergence, and model capacity, which can be quite intricate. We therefore refrain from making strong quantitative claims and instead rely on empirical validation to complement the theoretical results.
>
> ### Deterministic Sampling
> We agree that EqM’s gradient-based sampling is deterministic given a fixed initialization. The final sample is determined by the initial Gaussian noise, much like Flow Matching sampling with an ODE solver. In Appendix D.2, we include a short analysis clarifying the relationship between gradient-based sampling and standard ODE solvers under specific constraints on step size and number of steps.

---

### Official Review · Reviewer_sdot · 2025-10-31

**Soundness:** 2
**Presentation:** 2
**Contribution:** 2
**Rating:** 2
**Confidence:** 4

**Summary:**

The paper introduces Equilibrium Matching (EqM), a generative modeling framework that aims to learn a single, time-invariant equilibrium gradient of an implicit energy landscape. This contrasts with diffusion and flow models, which learn time-conditional, non-equilibrium dynamics. The method modifies the Flow Matching objective by introducing a magnitude function $c(\gamma)$ for the target gradient, constrained such that the gradient vanishes on the data manifold ($c(1)=0$). This allows sampling via optimization (e.g., gradient descent) on the learned landscape. The authors report a 1.90 FID on ImageNet 256x256, outperforming some existing diffusion and flow models. They also claim additional capabilities, such as denoising partially noised images, OOD detection, and image composition.

**Strengths:**

* **Generation Quality:** The model achieves an FID of 1.90 on class-conditional ImageNet 256x256, which is a good value.
* **Scalability:** The method demonstrates strong scaling properties. Figure 3 shows that EqM consistently outperforms the Flow Matching counterpart across different training epoch counts, model parameter counts, and patch sizes.
* **Inference Flexibility:** The optimization-based sampling framework is flexible. It allows for the use of standard optimization techniques, such as Nesterov Accelerated Gradient (NAG-GD), which improves FID from 1.93 (GD) to 1.90 (NAG-GD) on the EqM-XL/2 model  It also supports adaptive compute, which can reduce function evaluations by terminating based on gradient norm.
* **Partial Denoising:** A direct consequence of removing time/noise conditioning is the ability to denoise partially noised inputs without needing to be told the noise level. Figure 10 shows EqM's performance improves as the starting noise level decreases, whereas the baseline SiT model fails.

**Weaknesses:**

* **Incremental Novelty:** The core technical contribution is the modification of the noise-unconditional Flow Matching objective ($L_{uncond-FM}$, Eq. 2) to the EqM objective ($L_{EqM}$, Eq. 3). This change is driven entirely by multiplying the target gradient by a scalar function $c(\gamma)$ with the constraint $c(1)=0$. This seems like a minor modification to claim a new "framework."
* **Weak Empirical Link to EBMs:** The paper heavily motivates the method from an energy-based perspective. However, the explicit energy variants (EqM-E), which would validate this link, perform very poorly . Table 5 shows the "dot product" and "$L_2$ norm" variants yield FIDs of 73.40 and 75.53, respectively, far worse than the 57.54 of the implicit model ("none"). This undermines the claim that the model is effectively learning a useful, explicit energy landscape.
* **Contradictory Results and Excuses:** The method's performance is not consistent. On CIFAR-10, EqM (3.36 FID) is significantly worse than standard Flow Matching (2.09 FID) and only marginally better than noise-unconditional FM (3.96 FID). The authors' justification that the baseline was "extensively optimized" and noise schedules were "carefully selected" is a weak defense; a fundamentally superior method should not be so easily defeated by tuning.
* **Misleading "Unique Properties":**
    * The OOD detection capability (Table 6) relies on the "dot product" EqM-E variant. As established, this variant has poor generative performance (Table 5). This is a critical trade-off that is not properly acknowledged.
    * Image composition  is a known, standard property of EBMs. Presenting this as a "unique property" of EqM is misleading; it is merely an expected feature of the model class EqM *claims* to belong to.
* **Gap Between Theory and Practice:** The theoretical analysis (Statements 1 and 2) is used to justify that the model learns the data manifold. However, the derivations in Appendix C.1 and C.2 appear to assume a finite, discrete set of training points $\mathcal{X}$, not a continuous manifold.  Furthermore, there is no proof of either i) the fact that the gradient is zero ONLY at the points of interest and ii) that a sampling scheme will cover equally the local minima (is the data distribution recovered?). Finally, the convergence of GD like schemes is a well known result, it is not a contribution by the authors.
* **Contradictory Caption:** The caption for Table 5 states "EqM-E performs the best" The data in the table and the text in the "Energy Formulations" paragraph (which states "both formulations degrade performance") directly contradict this.

**Questions:**

* The theoretical derivations in Appendix C.1 and C.2 rely on a finite dataset $\mathcal{X}$. How do these proofs and "Statements" 1 and 2 extend from a finite set of points to the "data manifold"?
*  Why does the method's performance collapse relative to standard Flow Matching on CIFAR-10 (Table 8)? Blaming baseline optimization is insufficient. Does this imply EqM is only effective when paired with a transformer backbone (SiT) or on large-scale datasets, and not a universally better approach?
* Given that the explicit energy models (EqM-E) perform so poorly (Table 5), and the OOD detection relies on these models, isn't it more accurate to say that EqM forces a trade-off: high-quality generation (implicit model) *or* OOD detection (explicit model), but not both?
*  How is the $L_{EqM}$ objective (Eq. 3) fundamentally different from a noise-unconditional Flow Matching model (Eq. 2) that simply learns a re-scaled velocity field? The improvement from 40.81 FID (uncond-FM) to 32.85 FID (EqM) in Table 4 seems incremental. Why does this simple $c(\gamma)$ scaling constitute a new "framework"?

---

> ### Author Response · Authors · 2025-11-20
>
> ### Incremental Novelty
> We thank the reviewer for the thoughtful comments. Our main contribution is to show that **removing time conditioning from diffusion/flow models is highly beneficial when paired with an equilibrium structure** whose gradient vanishes on the data manifold. This enables gradient-based sampling with variable step sizes, adaptive compute, and flexible procedures such as composition or starting from partially noised images. We respectfully emphasize that **simplicity does not preclude novelty**: enforcing equilibrium-compatible gradients after removing time conditioning leads to both performance gains and new capabilities.
>
> ### Link to EBMs and Role of EqM-E
> Our EBM motivation comes from learning a **single equilibrium field**: EBMs assign low energy to data and high energy elsewhere, which EqM implements implicitly. EqM is thus an *implicit EBM*: it learns an energy landscape up to an additive constant and uses its gradient as a score field, without requiring explicit energies everywhere. We include an energy evolution plot (Appendix B.3, Fig. 12, https://ibb.co/R4DDMkXm) showing energy consistently decreases during sampling, supporting this link.
>
> We have renamed “Unique Properties” to “Properties of Equilibrium Matching” and clearly state that EqM-E underperforms implicit EqM. EqM-E is introduced only as a minimal explicit parameterization to probe the learned landscape and explore OOD scoring, not as a core contribution. We soften any “best of both worlds” language and clarify that current EqM-E results reflect a rudimentary explicit design, not a fundamental trade-off between generation and OOD detection.
>
> ### CIFAR-10 Results and FM-Specific Tricks
> The apparent “contradiction” on CIFAR-10 is largely due to **CIFAR-10 FM-specific training tricks** (EDM schedule and skewed timesteps) used for the FM baseline but not well-matched to EqM [1,2]. These tricks are used in the CIFAR-10 FM baseline only and not a common practice in ImageNet. We find:
>
> 1. These FM-specific tricks on CIFAR-10 do **not** transfer well to EqM.
> 2. When they are removed, **EqM outperforms FM**:
>
> |              | FM   | EqM  |
> |--------------|------|------|
> | w/ tricks    | 2.09 | 3.36 |
> | w/o tricks   | 3.70 | 3.32 |
>
> Without these enhancements, EqM is consistently better on CIFAR-10, in line with our ImageNet 256×256 results. We believe that there could be a similar set of tricks that can help EqM, similar to how EDM scheduling helps FM. We have updated Table 8 and Appendix B.1 accordingly.
>
> ### Theory, Practice, and Data Manifolds
> Our theoretical results provide conceptual guarantees under standard smoothness and regularity assumptions used in the generative modeling literature [3–5]. We explicitly acknowledge that training error and finite data limit direct applicability; the goal is to explain why equilibrium gradient fields are desirable and then validate EqM empirically on realistic datasets.
>
> Regarding data manifolds, the statements extend when the intrinsic manifold dimension is much smaller than the ambient space, though a full manifold treatment would require additional geometric assumptions and is beyond the current scope of the paper. In addition, the finite training set is what models actually see.
>
> ### Difference from Noise-Unconditional Flow Matching
> EqM differs from noise-unconditional FM in that:
> 1. It is **explicitly derived from an EBM perspective**, which prior noise-unconditional FM works do not emphasize.
> 2. It enforces a **vanishing-gradient condition** so that $\nabla E(x)=0$ on the data manifold, enabling standard gradient-descent sampling and robustness to step-size changes.
> 3. Its modified objective leads to improved FID on ImageNet 256×256 over strong FM baselines and enables generation from partially noised images without explicit timesteps.
>
> Taken together, the EBM-based interpretation, equilibrium objective, and empirical gains support EqM as a distinct equilibrium framework rather than a minor FM variant.
>
> [1] Lipman et al., *Flow Matching guide and code*, arXiv:2412.06264.
> [2] Karras et al., *Elucidating the design space of diffusion-based generative models*, NeurIPS 2022.
> [3] Chen et al., *Sampling is as easy as learning the score*, ICLR 2023.
> [4] Chen, Lee, Lu, *Improved Analysis of Score-based Generative Modeling*, ICML 2023.
> [5] Lee, Lu, Tan, *Convergence of Score-based Generative Modeling for General Data Distributions*, ALT 2023.

---

### Official Review · Reviewer_gTDd · 2025-11-01

**Soundness:** 2
**Presentation:** 3
**Contribution:** 2
**Rating:** 4
**Confidence:** 5

**Summary:**

The paper introduces a generative modeling framework that replaces the non-equilibrium, time-conditioned dynamics of diffusion and flow models with a single equilibrium gradient derived from an implicit energy landscape. Instead of simulating a time-dependent reverse process, EqM performs optimization-based sampling at inference time using gradient descent on this learned energy surface, allowing adaptive step sizes and computational efficiency. Class-conditional generation achieves an FID of 1.90 on ImageNet 256×256, surpassing diffusion and flow baselines. It also supports tasks such as denoising, OOD detection, and image composition without architectural changes. Conceptually, EqM aims to unify flow-based and energy-based generative modeling through equilibrium dynamics, offering a simpler alternative to time-dependent generative models.

**Strengths:**

1) The writing is clear, the abstract is concise and sets up the new paradigm (equilibrium vs non-equilibrium) nicely.

2) Moving away from time-conditional flows/diffusions to a single equilibrium landscape is a compelling conceptual shift.

3) The reported generation quality (e.g., FID 1.90 on ImageNet 256×256) is extremely competitive, if accurate.

4) List of downstream tasks (denoising, OOD detection, image composition) shows the authors have thought about broader applicability.

**Weaknesses:**

1) While the equilibrium viewpoint is interesting, many past works have considered implicit energy landscapes and hybrid flows/EBMs, to name a few, Energy Matching [1], VAPO [2] and Action Matching [3]. Authors need to more explicitly compare and contrast with such prior art, clarify what is distinct in their equation/learning/training paradigm.

2) The connection between the noise/time-unconditional FM (2) and the noise/time-conditioned FM (1) remains unclear. Does (1) converges to (2) is some time/noise limit? Or is (2) some generalization of (1) in terms of the sampling path or training objective formulation?

3) The theoretical section somewhat high-level. It does not answer questions like: what assumptions guarantee that the learned equilibrium landscape corresponds to the data manifold? How many gradient steps are required?

4) No evidence of ablations that test sensitivity to sampling step-size, optimizer type, landscape smoothness, number of gradient steps, or compare optimisation-based sampling vs more standard sampling (e.g., Langevin dynamics, flow inversion). These are crucial for understanding where EqM gains come from.

5) It would be helpful to see failure cases: e.g., where the energy landscape optimisation fails (mode collapse, slow convergence, sensitivity to initialization).

**Questions:**

Please consider addressing the weaknesses noted above.

---

> ### Author Response · Authors · 2025-11-20
>
> ### Relation with Past Works
> We thank the reviewer for the helpful suggestion to clarify the relationship between EqM and prior methods such as Energy Matching, VAPO, and Action Matching. EqM differs fundamentally in that it removes all time/noise conditioning and directly learns a **stationary equilibrium gradient field** satisfying $\nabla E(x)=0$ on the data manifold. This equilibrium constraint is absent from prior formulations. In contrast to trajectory-based matching methods (e.g., VAPO, Action Matching), EqM learns a global potential whose gradient defines a unified generative field, rather than fitting individual trajectories. We have updated the Related Works section to include this comparison.
>
> | method          | time conditioning | objective                                       |
> |-----------------|-------------------|-------------------------------------------------|
> | EqM (ours)      | no                | a single equilibrium energy landscape \(E(x)\)  |
> | Energy Matching | yes               | energies or scores of a reference diffusion/EBM |
> | VAPO            | yes               | variational objectives over time-dependent trajectories |
> | Action Matching | yes               | actions of a reference flow                     |
>
>
>
> ### Connection Between Equations
> We appreciate the request for a clearer connection between the equations. Eq. 2 is a time-unconditional variant of the Flow Matching algorithm (Eq. 1) proposed by [1]. It removes the time input to the model entirely, while preserving all other components of Flow Matching training. This yields a time-unconditional model that still follows the Flow Matching framework but is explicitly designed for equilibrium dynamics.
>
> ### Theoretical Section
> We thank the reviewer for engaging with the theoretical section. We have clarified the theoretical section to emphasize the takeaways. Statements 1 and 2 together ensure that the learned landscape converges to the data manifold, while Statement 3 guarantees a standard convergence rate for gradient descent on this landscape. Taken together, they provide a coherent theoretical picture: the energy landscape is shaped so that its critical points coincide with the data manifold, and standard optimization guarantees apply to sampling. We have modified our Analysis section to reflect these changes.
>
> ### Sensitivity at Sampling
> We evaluated EqM’s sensitivity to sampling hyperparameters through several experiments:
>
> - **Step size:** Figure 7 (https://ibb.co/Txxw0Kk9) reports performance under different sampling step sizes and shows that EqM is substantially less sensitive to the step size than standard Flow Matching.
> - **Optimizers and samplers:** Table 2 (shown below) reports results for a range of optimizers and samplers, including both gradient-based optimizers and integration-based samplers. All tested EqM variants outperform the FM baseline.
> | model          | sampler                | $\eta$  | $\mu$ | FID          |
> |----------------|------------------------|---------|-------|--------------|
> | SiT-XL/2       | Euler {\scriptsize (ODE)} | 0.0040  | -     | 2.10         |
> | SiT-XL/2       | Heun {\scriptsize (SDE)}  | 0.0040  | -     | 2.06         |
> | EqM-XL/2       | Euler {\scriptsize (ODE)} | 0.0017  | -     | 1.93         |
> | EqM-XL/2       | GD                     | 0.0017  | -     | 1.93         |
> | **EqM-XL/2**   | NAG-GD                 | 0.0017  | 0.3   | **1.90**     |
>
> - **Number of steps:** Figure 6 (https://ibb.co/jP5vLqwz) shows performance under different numbers of sampling steps.
>
>
>
>
>
> We have performed additional experiments using the Adam optimizer, and observed decent generation quality:
> | sampler                 | GD | Adam |
> |-------------------------|-----------|-----------|
> | FID                     | 32.85      | 36.35       |
>
> Together, these results support our claim that EqM yields a more robust sampling procedure. Additionally, we have performed new experiments on the Adam optimizer, demonstrating that second-order optimizers can also function properly on EqM. We have modified the Appendix to include these results (Table 12).
>
> ### Failure Cases
> We agree that it is important to discuss failure modes. One such case is the EqM-E variant, which constrains the model to output a single scalar energy value. We observe a drop in generation quality in this setting, which we hypothesize is due to optimization difficulties arising from backpropagating through this scalar energy value. This further motivates the implicit EqM formulation used in the main experiments. We have modified our discussion on EqM-E to reflect this response.
>
> [1] Sun, Q., Jiang, Z., Zhao, H. and He, K., 2025. *Is Noise Conditioning Necessary for Denoising Generative Models?* arXiv preprint arXiv:2502.13129.

---

### Author Response · Authors · 2025-12-01

We thank the AC and reviewers for their time and their careful and constructive feedback. We are glad that reviewers find **Equilibrium Matching (EqM)** conceptually novel (gTDd, xen9, Ct1d), extremely strong in empirical performance (gTDd, sdot, xen9, Ct1d), and broadly applicable as a simple equilibrium framework for generative modeling (gTDd, sdot). Reviewers also appreciated that the paper is clearly written and well structured, and that the empirical results on ImageNet are particularly compelling.

However, reviewers raised several important concerns. In particular, they asked for:
1. clearer **comparisons to prior EBM-related and equilibrium / hybrid flow methods**, including Energy Matching, VAPO, and Action Matching (gTDd, sdot, xen9);
2. a more precise characterization of the **strength and scope of the theoretical guarantees**, and how they connect to practical large-scale training (gTDd, sdot, xen9, Ct1d);
3. stronger **empirical validation beyond ImageNet**, especially on CIFAR-10, more systematic comparisons to EBMs/Energy Matching, and concrete evidence about **sampling efficiency and robustness** of optimization-based sampling (gTDd, sdot, xen9, Ct1d);
4. clarification of the **role of EqM-E** (the explicit energy variant), including its weaker FID and its use for OOD detection and “unique” equilibrium properties (sdot, xen9).

We believe that we have been able to address these concerns through additional clarifications and experiments, together with per-reviewer replies.

---

## 1. Clarifications

### 1.1. Scope and goals of EqM.

Some reviewers questioned whether EqM is primarily a new **equilibrium/EBM formulation** or just a modest variant of Flow Matching. We clarify that our main goal is to show that **removing time/noise conditioning and enforcing an equilibrium condition**, and that the learned potential field has vanishing gradient on the data manifold, yields a practically useful alternative to time-conditioned diffusion/flow models. EqM is designed to:

- learn a **single, time-unconditional equilibrium gradient field** instead of a trajectory-conditioned velocity field;
- support **optimization-based sampling** directly in data space, enabling partial denoising and compositional updates;
- remain compatible with modern high-performance architectures (e.g., SiT) and large-scale training on ImageNet.

Our focus is therefore on establishing EqM as a **simple but non-trivial equilibrium framework** with strong empirical performance and practical benefits.

### 1.2. Relation to prior equilibrium / hybrid methods and noise-unconditional FM.

A central concern was how EqM differs from prior work such as **Energy Matching, VAPO, and Action Matching**, as well as from a “time-unconditional” Flow Matching baseline. In the revision, we:

- add a **structured comparison** that contrasts EqM with these methods along axes such as **time/noise conditioning**, **objective**, and **training target**.

| method          | time conditioning | objective                                       |
|-----------------|-------------------|-------------------------------------------------|
| EqM (ours)      | no                | a single equilibrium energy landscape \(E(x)\)  |
| Energy Matching | yes               | energies or scores of a reference diffusion/EBM |
| VAPO            | yes               | variational objectives over time-dependent trajectories |
| Action Matching | yes               | actions of a reference flow                     |

| model  | FM   | IGEBM | Energy-based U-Net | Energy Matching | EqM  |
|--------------------|------|-------|--------------------|-----------------|------|
| FID| 3.70 | 37.9  | 6.8| 3.34   | 3.32 |

- emphasize that Energy Matching and related methods typically **retain time/noise conditioning**, or match energies along specific trajectories, whereas EqM directly learns a **global equilibrium field** whose gradient vanishes on the data manifold.
- clarify that Eq. (2) in our paper should be seen as a **time-unconditional variant of Flow Matching**, but EqM further imposes the equilibrium constraint and interprets the learned field as the gradient of an implicit energy, leading to different sampling behavior and empirical properties.


We have updated the related-work and method sections to make these conceptual and technical differences explicit.

---

> ### Author Response · Authors · 2025-12-01
>
> ### 1.3. Theoretical guarantees and their scope.
>
> Several reviewers requested a clearer statement of what our theoretical results do and do not guarantee. In the rebuttal, we:
>
> - reframe our theoretical results as **conceptual guarantees under standard smoothness and regularity assumptions** commonly used in the diffusion/EBM literature, rather than as exact finite-data theorems.
> - clarify that the statements about **critical points of the learned energy aligning with the data manifold** and the **gradient-descent convergence rate** are meant to explain *why* equilibrium learning can support optimization-based sampling, not to fully characterize the finite-sample behavior of large models.
> - explicitly discuss how training error and model misspecification can lead to approximate rather than exact satisfaction of the equilibrium conditions, and why we therefore lean on **empirical validation** to support our claims.
>
> We have adjusted the wording to avoid overstating the formal strength of the theory while preserving its explanatory role.
>
> ### 1.4. Role of EqM-E, OOD detection, and “unique” properties.
>
> Reviewer sdot in particular worried that the OOD and explicit-energy experiments rely on **EqM-E**, whose FID is weaker, and that our discussion of “unique” equilibrium properties might be misleading. In response, we:
>
> - highlight that EqM-E’s **worse FID** is expected due to optimization constraints through a scalar energy.
> - soften the language around “unique properties” and present partial denoising and compositional updates as **natural consequences of equilibrium-based optimization**, which are shared with other EBMs, while emphasizing that EqM shows these properties in a **strong ImageNet regime** with modern architectures.
>
> These clarifications are reflected both in the main text and in the discussion of OOD and composition experiments.
>
> ---
>
> ## 2. Additional experiments
> *(please see per-reviewer responses for more reviewer-specific experiments)*
>
> ### 2.1. CIFAR-10 and comparisons to EBMs / Energy Matching.
>
> Reviewers requested stronger evidence that EqM’s benefits extend beyond ImageNet and beyond comparisons only to Flow Matching. In the rebuttal and revision, we:
>
> - Add **CIFAR-10 experiments** that carefully separate Flow Matching trained with and without EDM-style timestep tricks and skewed schedules. We show that:
>   - With FM-specific tuning, FM achieves better FID than EqM;
>   - Without those tricks, EqM **beats** FM on CIFAR-10, aligning with the ImageNet observations.
>
> |              | FM   | EqM  |
> |--------------|------|------|
> | w/ tricks    | 2.09 | 3.36 |
> | w/o tricks   | 3.70 | 3.32 |
> - Include new **comparisons to EBMs and Energy Matching** on CIFAR-10, showing that EqM attains **the best FID among all tested EBM-related baselines**, thereby strengthening the claim that EqM is competitive not only with flow-based models but also with strong EBMs.
>
> | model  | FM   | IGEBM | Energy-based U-Net | Energy Matching | EqM  |
> |--------------------|------|-------|--------------------|-----------------|------|
> | FID| 3.70 | 37.9  | 6.8| 3.34   | 3.32 |
>
> These results are summarized in the main paper and detailed in the appendix.
> ### 2.2. Sampling efficiency and dynamics.
>
> Concerns about the **efficiency** of optimization-based sampling led us to add more detailed timing and dynamics analyses:
>
> - We report **wall-clock time per batch** for EqM and Flow Matching under matched step counts. The results show that EqM achieves **strictly better FID while being faster** than the FM baseline in the reported ImageNet setting.
>
>
> | model                   | FM        | FM        | EqM    | EqM     |
> |-------------------------|-----------|-----------|--------|---------|
> | sampler                 | Euler ODE | Heun SDE  | GD     | NAG-GD  |
> | sampling steps          | 250       | 250       | 250    | 250     |
> |wall-clock time|   70s | 272s | 59s | 55s |
> | FID                     | 2.10      | 2.06      | 1.93   | 1.90    |
>
> - We add **energy-evolution plot** (https://ibb.co/R4DDMkXm) that tracks the energy along sampling trajectories, showing that optimization-based sampling monotonically decreases the learned energy, providing empirical evidence that the gradient field behaves as an effective descent direction on the learned landscape.
>
> Together, these additions support our claim that EqM provides **efficient and practical optimization-based sampling**, not just a conceptual possibility.

---

> > ### Author Response · Authors · 2025-12-01
> >
> > ### 2.3. Sensitivity analyses and optimizers.
> >
> > Reviewers asked how sensitive EqM is to sampling hyperparameters and whether more advanced optimizers could further improve performance. To address this, we:
> >
> > - clarify existing **ablations** over step size, number of gradient steps, and choice of optimizer. These results show that EqM’s FID is **robust over a reasonable range of hyperparameters**.
> > - include explicit **Adam-based sampling experiments**, which demonstrates that Adam optimizer works reasonably well. We discuss the potential of more sophisticated optimization (including second-order methods) as promising future work rather than as a requirement for good performance.
> >
> > | sampler                 | GD | Adam |
> > |-------------------------|-----------|-----------|
> > | FID                     | 32.85      | 36.35       |
> >
> > These experiments, along with the timing results, show that EqM does not rely on fragile hyperparameter choices to be effective.
> >
> >
> > ---
> >
> > We have incorporated these changes in our revision and carefully polished the writing to avoid potential misunderstandings regarding novelty, theoretical scope, and empirical claims. We are thankful for the opportunity to clarify and improve the paper.

---

### Meta-Review · Area_Chair_gdtp · 2025-12-19

**Summary:**

Despite the strong empirical results on ImageNet, the consensus on rejection is driven by concerns regarding the fundamental novelty and theoretical grounding of the proposed method. Reviewers sdot and gTDd pointed out that the technical contribution is largely incremental, effectively amounting to a noise-unconditional Flow Matching objective with a specific gradient scaling, rather than a distinct "equilibrium" framework. Furthermore, the theoretical analysis remains weak; it relies on idealized "perfect training" assumptions that do not hold in practice, and the authors failed to provide rigorous guarantees for finite-data regimes. Crucially, the "explicit energy" variant (EqM-E) performs poorly, which significantly undermines the central claim that the model is learning a valid energy landscape, suggesting instead that it functions merely as a specialized vector field estimator without the benefits of true energy-based modeling.

**Reviewer Concerns:**

The authors addressed concerns regarding sampling efficiency and empirical comparisons. In response to Reviewers xen9 and Ct1d, they provided concrete wall-clock benchmarks demonstrating that EqM is faster than the Flow Matching baseline while achieving better FID. Finally, the authors satisfied requests from Reviewers gTDd and sdot for broader context by adding structured comparisons to Energy Matching and VAPO, showing EqM outperforms these prior equilibrium-based methods.

However, fundamental concerns regarding theoretical rigor and the validity of the energy-based formulation remain. Reviewers sdot and Ct1d noted that the theoretical proofs rely on "perfect training" and infinite data assumptions, which the authors acknowledged as "conceptual guarantees" rather than rigorous finite-sample results. Furthermore, the significant performance degradation of the explicit energy variant (EqM-E) remains a critical weakness highlighted by Reviewer sdot; this failure undermines the central claim that the framework truly learns a valid energy landscape, suggesting it functions more as a specialized vector field estimator.

**Reviewer Scores:**

Reviewer gTDd: While the authors added the requested comparisons to Energy Matching and VAPO, the reviewer’s underlying concern about the "fair" soundness and contribution likely persists. The rebuttal confirmed that the theoretical claims are "conceptual" rather than rigorous, which validates the reviewer's initial hesitation regarding the theoretical grounding of the equilibrium landscape compared to established flow-based methods.

Reviewer sdot: The authors' admission that the explicit energy variant (EqM-E) performs poorly directly validated this reviewer's most critical point: that the method fails to bridge the gap to Energy-Based Models effectively. Furthermore, the explanation regarding CIFAR-10 performance likely reinforced the reviewer’s view.

Reviewer xen9: They were already convinced of the novelty and soundness of the equilibrium approach. The rebuttal’s inclusion of wall-clock timing and energy evolution plots only served to confirm their positive assessment regarding sampling efficiency, leaving them with no reason to lower their score.

Reviewer Ct1d: This reviewer would likely keep their score at 4. Although the authors provided the requested wall-clock comparisons, the rebuttal's concession that the theoretical analysis relies on "perfect training" assumptions likely failed to alleviate the reviewer's primary concern about soundness.

---

### Decision · Program_Chairs · 2026-01-26

Reject